# Predicting Bovine Respiratory Disease Risk in Feedlot Cattle in the First 45 Days Post Arrival

**DOI:** 10.3390/pathogens11040442

**Published:** 2022-04-06

**Authors:** Hector A. Rojas, Brad J. White, David E. Amrine, Robert L. Larson

**Affiliations:** Beef Cattle Institute, Kansas State University, Manhattan, KS 66506, USA; hectorrojas@vet.kssu.edu (H.A.R.); damrine@vet.ksu.edu (D.E.A.); rlarson@vet.ksu.edu (R.L.L.)

**Keywords:** bovine respiratory disease, predictive modeling, economic analysis

## Abstract

Bovine respiratory disease (BRD) is the leading cause of morbidity in feedlot cattle. The ability to accurately identify the expected BRD risk of cattle would allow managers to detect high-risk animals more frequently. Five classification models were built and evaluated towards predicting the expected BRD risk (high/low) of feedlot cattle within the first 45 days on feed (DOF) and incorporate an economic analysis to determine the potential health cost advantage when using a predictive model compared with standard methods. Retrospective data from 10 U.S. feedlots containing 1733 cohorts representing 188,188 cattle with known health outcomes were classified into high- (≥15% BRD morbidity) or low- (<15%) BRD risk in the first 45 DOF. Area under the curve was calculated from the test dataset for each model and ranged from 0.682 to 0.789. The economic performance for each model was dependent on the true proportion of high-risk cohorts in the population. The decision tree model displayed a greater potential economic advantage compared with standard procedures when the proportion of high-risk cohorts was ≤45%. Results illustrate that predictive models may be useful at delineating cattle as high or low risk for disease and may provide economic value relative to standard methods.

## 1. Introduction

Bovine respiratory disease (BRD) remains the costliest disease in the American feedlot industry costing between approximately $800 and $900 million annually [1]. Cattle arriving to a feedlot are managed in groups and management decisions such as the administration of antimicrobial metaphylaxis upon arrival is determined on the perceived risk of a high percentage of cattle within a group developing BRD [2]. Feedlot decision makers use BRD risk classification of high risk versus low risk to decide whether cohorts of cattle will or will not receive metaphylactic treatment upon arrival to the feedlot. Many factors can drive the perceived BRD risk of incoming cattle groups which can vary by personnel and organizational policies. As a result, misclassifications may occur when determining BRD risk classes. These misclassifications may negatively impact cattle health and ultimately lead to increased expenditures towards treating health-related events.

Previous work has encouraged the use of operational feedlot data to predict BRD health outcomes [3,4]. The ability to use feedlot data available at the time cattle arrive to correctly predict and classify incoming cohorts of cattle into high- or low-risk groups would allow for more judicial use of antimicrobials, potentially increase economic performance, and allow personnel to focus their efforts on those cohorts expected to classify as high risk. Previously investigated cohort risk factors such as average body weight, sex, quarter of arrival, and cohort size have all been found to be associated with BRD morbidity risk [5,6,7]. Pen housing conditions such as pen area and bunk space per head have also been explored as risk factors for BRD [8]. The objective for this study was to assess the ability of five different classification algorithms to accurately predict an incoming group of cattle’s risk classification (high/low) using commercial feedlot data during the first 45 days on feed (DOF). In addition, the economic performance of each model was evaluated to determine if a potential health-cost advantage was present with the use of each models’ final model outputs.

## 2. Results

### 2.1. Descriptive Statistics

The study population data included 1733 distinct cohorts of cattle representing 188,118 individual animals. Each cohort was classified into a risk class category of high or low based on BRD morbidity risk in the first 45 DOF in relation to a 15% cutoff. In our training dataset there were 141 (10.85%) cohorts that were high risk and 1159 (89.15%) cohorts that were low risk using the 15% cutoff. In the test dataset there were 47 cohorts that were high risk (10.85%) and 386 cohorts that were low risk (89.15%). The mean BRD morbidity in the first 45 days for all cohorts of cattle was 6.30%. The mean BRD morbidity in the first 45 DOF for high-risk cattle was 27.07% and the mean morbidity for low-risk cattle was 3.77%. A total of 1300 cohorts were partitioned in the training dataset, while 433 cohorts were partitioned in the test dataset. The prevalence of high-risk cohorts in the training and test dataset was 10.85%, respectively.

### 2.2. Model Performance Diagnostics

#### 2.2.1. Area under the Curve and Classification Accuracy

The accuracies and area under the curve (AUC) of the five classification models were evaluated using the test dataset (Table 1). AUC was calculated using each model generated ROC curve from the test dataset. AUC of the models ranged from 0.682 to 0.789 for decision tree and random forest, respectively. Accuracy of the models ranged from 10.9% to 79.4% for naïve Bayes and logistic regression, respectively.

#### 2.2.2. Sensitivity and Specificity

Sensitivity between the classification models ranged from 44.7% to 100% (Table 1). The highest sensitivity, 100%, was achieved using the naïve Bayes model. The lowest sensitivity, 44.7%, was achieved using the decision tree model. The model with the highest specificity was the decision tree model at 83.7%. The model with the lowest specificity was the naïve Bayes model at 0%.

#### 2.2.3. Positive/Negative Predictive Value

The model positive predictive value and negative predictive value with a 10.85% prevalence of high-risk cohorts ranged from 10.9% to 25.0% and 92.6% to 97.9%, respectively (Table 1). The model with the highest positive predictive value was the decision tree model (25.0%); the model with the lowest positive predictive value was the naïve Bays (10.9%). The model with the highest negative predictive value was the random forest (97.9%), whereas the model with the lowest negative predictive value was the decision tree (92.6%).

### 2.3. Economic Results

The derived sensitivity and specificity for the control scenario was 83.75% and 59.79%, respectively. The Net Health Cost Benefit and the difference from the control ($/animal) were calculated for each of the five models. The difference from the control for each model was variable depending on the proportion of high-risk cohorts to low-risk cohorts in the population. As a result, the potential economic advantage/disadvantage of using a model compared to not using a model (control) was volatile at different proportions of high-risk cohorts to low-risk cohorts (Figure 1). In our study, logistic regression and random forest models always offered a positive, but small, difference (higher $/animal) from the control method across all possible prevalence of high-risk cohorts as they had a higher cost per head advantage at all proportions of high-risk cohorts to low-risk cohorts. Decision tree models had a positive difference from the control when the proportion of high-risk cohorts to low-risk cohorts are below 45%. Naïve Bayes models had a positive difference from the control when the proportion of high-risk cohorts to low-risk cohorts was above approximately 83%. Linear discriminant models had a positive difference from the control when the proportion of high-risk cohorts was above approximately 25%.

## 3. Discussion

Protocols related to health management for cattle entering a feedlot are often based on the expected risk class of disease within the group. Accurately predicting the health outcome of incoming cohorts of cattle can serve to increase feedlot performance, efficiency, and economic performance. Previous studies have incorporated cohort characteristics and risk factors at arrival into predictive algorithms to accurately classify cohorts into classes related to BRD morbidity [4,9]. However, these studies did not incorporate variables linked to pen housing conditions such as pen and bunk space that previously have been investigated [8]. In this study we evaluated the diagnostic ability of five predictive algorithms to predict BRD morbidity risk (high or low) for cattle arriving to a feedlot within the first 45 days post arrival using a 15% cutoff while incorporating previously associated risk factors for BRD. We evaluated the predictive ability towards the outcome of interest of high/low BRD morbidity risk in the first 45 DOF from the models that were produced from the use of the variables in our analysis. The amount of BRD morbidity risk within a cohort that is acceptable before management intervention differs among feedlot producers; therefore, an economic analysis was performed to better determine if the predictive performance of any of the five models would be economically beneficial compared with a person classifying incoming groups of cattle high or low risk.

The AUC of each model’s receiver operating characteristic (ROC) curve was the metric used to rank the predictive performance of each model. In general, an AUC of 0.5 describes a model that has no discriminatory ability and serves as a model that has a 50% probability of correctly classifying an observation. An AUC between 0.7 to 0.8 is acceptable, 0.8 to 0.9 is excellent, and more than 0.9 is outstanding [10]. In our study, the model with the highest AUC is the random forest model at 0.79, with the range of AUC being 0.68 to 0.79 between all models. This indicates the models’ performance ranged from poor to acceptable based on AUC. The overall accuracy of each model was also calculated; however, evaluating accuracy alone may be misleading when interpreting the final results in an imbalanced dataset [11]. This was because the majority of the cohorts in the test data were classified in the low-risk category (<15% were treated for BRD during the first 45 DOF). If a predictive model classified every cohort as low risk it would have an accuracy of 89.1%. On the surface, this appears to be an acceptable accuracy; however, when a model does not have any discriminating ability and predicts all cohorts the same, the model is relatively useless. As a result, AUC was used to rank the performance of each model as it avoids this bias and allows us to better understand the predictive ability of our models.

The prevalence of high-risk cohorts in our dataset was 10.85% (188 high-risk cohorts out of 1733 total cohorts). The PPV represents the proportion of predicted high-risk cohorts that were truly high risk. The negative predictive values report the proportion of predicted low-risk cohorts that are truly low risk. The PPV from our models ranged from 10.9% (naïve Bayes) to 25% (decision tree), demonstrating that our final models have a low probability of predicting positives (high risk) that are actual positives. In contrast, our models’ NPV ranged from 92.6% (decision tree) to 97.9% (random forest), with one algorithm generating a division by zero error (naïve Bayes). The naïve Bayes model had this error because the model’s specificity was 0%, so it predicted every cohort to be positive (TP or FP). Four of the five models created from this data perform well at predicting negatives (low risk) that are actual negatives. As prevalence decreases, positive predictive value will increase and the negative predictive value will decrease, and vice versa [12]. When evaluating the PPV and NPV for each model the expected prevalence of the outcome of interest should be considered.

Determining the practicality of each model’s usage in a feedlot setting requires both knowledge of cohort-level feedlot data characteristics and predictive modeling. Feedlot producers generally use expected BRD risk to decide to give an incoming group of cattle metaphylactic treatment to reduce cattle morbidity. Costs are associated with correctly/incorrectly administering metaphylaxis to incoming cohorts of cattle. The ability to utilize a predictive model to predict the expected BRD risk of an incoming cohort may aid feedlot operations in correctly administering metaphylaxis treatment and consequently maximize profitability. Previous studies have utilized predictive analytic techniques to attempt to predict an outcome of interest; however, these studies have not incorporated an economic component in their model analysis [3,4,9]. We included a deterministic economic approach to estimate a Net Health Cost Benefit cost for the use of each model when classifying cohorts into high- or low-BRD morbidity risk. As a deterministic approach was used to calculate costs, there is no randomness or variability in our results from potential factors that may impact the population. This was not meant to represent a full-scale economic analysis and only estimates the costs and potential benefits from associated treatments and use of metaphylaxis to mitigate BRD. Cohorts called high risk (positive) will receive metaphylaxis and cohorts identified as low risk (negative) will not receive metaphylactic treatment. The costs of a true negative, false negative, true positive, and false positive at different proportions of high-risk cohorts were calculated to determine the costs associated with each outcome. As a result, the calculations for these costs do not consider expenditures related to additional factors related to feed costs, management costs, and other potential costs that were not included.

The model that provided the greatest economic advantage was dependent on the prevalence of high-risk cohorts in the population and the severity of BRD morbidity risk within those high-risk groups. A lower prevalence of high-risk cohorts will favor models that have a higher specificity. In contrast, a greater prevalence of high-risk cohorts will favor models that have a higher sensitivity. Depending on the expected prevalence of disease, managers can determine which specific is the best option and whether it is more beneficial to use a model compared to not using a model.

In our study, sensitivity represents the model’s ability to correctly identify cohorts that are high risk based on a selected cutoff; specificity represents the model’s ability to correctly identify cohorts that are low risk on a selected cutoff. Determining which metric to prioritize is dependent on the importance of minimizing false positives or false negatives and the cost of each outcome. False positives with this data would be a cohort that was truly low risk (<15% BRD morbidity 45 DOF) yet was predicted to be high risk. False negatives would be a cohort that was truly high risk (≥15% BRD morbidity 45 DOF) yet was predicted to be low risk. The costs and consequences for each type of error are different. An increase in false positives may lead to additional unnecessary metaphylaxis treatment costs that are administered to cohorts that are at low expected risk for BRD. In our study, the cost of a false positive would be on average an extra $23.60 spent per animal for each false-positive cohort that did not need metaphylactic treatment. False negatives may lead to negative health outcomes as cattle would not receive metaphylaxis for respiratory disease when they truly needed treatment. This misclassification can lead to increased health costs, losses in performance, and potentially increased mortality. In our study, the cost of a false negative was the loss of value of a sick animal compared with a healthy animal, which was an estimated lost value of $151.18 per treated animal; the number of treated animals varies based on estimated morbidity.

The decision tree model has the highest potential economic advantage compared to other models when the proportion of high-risk cohorts present was ≤45%. For example, when 5% of the cohorts entering the feedlot are high-risk cohorts the decision tree model offers a potential $5/animal health cost advantage compared with human control. The decision tree model had the highest specificity, which increases the model’s ability to detect low-risk cohorts. Therefore, at this level of prevalence (5% high risk, 95% low risk) the increased specificity of the decision tree was more valuable ($5/animal health cost advantage) than the models with higher sensitivity. These results agree with a previous study that reported that increasing diagnostic test specificity increased economic net returns in comparison to increasing sensitivity [13]. However, in our study, the estimated health-cost advantage of the decision tree compared with the human control decreased as the proportion of high-risk cohorts increased. Once the prevalence of high-risk cohorts was 83% and above, the naïve Bayes model, which has the highest sensitivity, has an economic advantage over the control. However, at this level of high risk, the cohorts’ feedlot managers would likely not distinguish between high- and low-risk cohorts and would likely treat all groups with metaphylaxis well before the proportion reaches 83%. The cost of using each model was dependent on the prevalence of high-risk cohorts arriving to the feedlot, which should be considered when determining the health-cost advantage to using these models compared with a human.

A potential limitation may have been that the feedlots in our dataset only represented Midwest feedlots and the data may not represent cohorts of cattle from all feedlots in terms of the dates recorded, location, cattle types, and many other factors. Another limitation was that we did not have data indicating whether groups of cattle in our dataset had received metaphylactic treatment. This could have impacted our outcome of interest, BRD morbidity risk in the first 45 DOF, as we were not aware whether cohorts in either the low- or high-risk category were previously mass treated. This could have affected what expected risk cohorts were placed into at the 15% treatment cutoff as we do not know if the percent of cattle treated for BRD in each cohort was affected by metaphylaxis. This could potentially impact our results, by placing truly high-risk cattle into the incorrect classification. For example, if a truly high-risk group of cattle was identified by feedlot personnel and received metaphylactic treatment, then the overall percentage of cattle that were treated for BRD within the first 45 days on feed in this cohort could have potentially been lower than 15% due to metaphylaxis and they would not have been included in the high-risk category in our study. As a result, there are likely cattle that are truly high risk present in the low-risk category. The calculated sensitivity and specificity for our human control was representative of a subset of data that was available and does not represent the sensitivity and specificity for all feedlots. This was a small portion of data (*n* = 177 cohorts) and the generalizability of this data most likely does not reflect all feedlots.

## 4. Materials and Methods

### 4.1. Data

Retrospective data from 10 Midwest feedlots were collected between January 2018 and April 2020 and utilized for this study. A cohort was defined as a group of cattle purchased and managed in a similar manner and housed together throughout the study period during the initial 45 days on feed post arrival. Groups of cattle were procured by each feedyard and no information on prior management or health prevention procedures were available on the cattle cohorts. All data included in the study were collected on the animals at or after feedyard arrival. Cohort- and individual-level variables were included in the dataset. Cohort-level variables included: average arrival weight (total weight of all animals within the cohort divided by the total head in that cohort), number of cattle in cohort at arrival, arrival date quarter, and sex (steers, heifers, mixed gender). Individual-level data included the total number of individual first treatments in each cohort for BRD within the first 45 DOF. Bovine respiratory disease incidence, our outcome, was defined as the number of cattle that were treated at least once for BRD based on feedlot diagnosis within the first 45 DOF divided by the size of the cohort. The case definition for a BRD treatment was any animal that received an antimicrobial treatment for BRD during the first 45 DOF. Cases were limited to first BRD treatments only and any additional treatments were excluded from analysis. If an animal was treated more than once, the first treatment record was utilized. Cohorts with missing data for any of these variables were excluded from the study population.

Pen housing variables were calculated for each cohort, including: pen area (sq. m), bunk space available (m), pen area per head (sq. m), and bunk space per head (m). Dimensions of each pen were measured utilizing the ‘ruler tool’ Google Earth Pro [14]. Pen area was calculated by measuring the square meters of each pen. This was done by multiplying the length of the pen by the width of the pen if the pen shape was square or rectangular. If the pen had an irregular polygonal shape, then the ‘polygon tool’ was utilized to measure the area of the geometric shape of the pen. Linear bunk space was recorded by measuring the length (m) of visible bunk in each pen. Pen area per head was calculated by dividing pen area (sq. m) by the cohort size at arrival for each cohort. Bunk space per head was calculated by dividing pen bunk space available (m) by cohort size at arrival for each individual cohort. Cohorts without available pen housing measurements were removed from the dataset. Cohorts that were housed in 2 or fewer pens within the first 45 DOF were included for analysis. If a cohort was housed in 1 pen for the entirety of the 45 DOF period then the dimensions of the 1 pen were used for analysis. If a cohort was housed in 2 pens during the 45 DOF period then the dimensions of the second pen were used for analysis, but only when the cohort was limited to <7 DOF in the first pen. Any cohorts that were moved between 3 or more pens during the first 45 DOF were excluded from analysis.

### 4.2. Data Preparation

The cumulative percent of cattle receiving a first treatment for BRD within the first 45 DOF was calculated for each cohort. The primary study outcome was expected cohort-level BRD risk classification (high or low risk) based on a treatment cutoff of 15% total BRD morbidity within the first 45 days on feed that has previously been used in prior research [13]. If 15% or more animals in a cohort were treated for BRD at least once in the first 45 DOF, the cohort was classified as a high-risk cohort. If less than 15% were treated for BRD during the first 45 DOF, then the cohort was classified as a low-risk cohort. A new binary cohort-level variable was created to represent the cutoff and populated with a value of 1 if BRD morbidity was greater than or equal to 15% or 0 if BRD morbidity was less than 15%.

### 4.3. Data Partitioning

Models may become overfitted and provide inaccurate biased estimates when utilizing a single dataset for training and testing the models. An overfitted model developed with a single dataset may fail to predict new data sets accurately [15]. Multiple datasets are used to avoid biased estimates and improve each model’s discrimination ability by evaluating final diagnostic performance in a dataset independent of data used for model building phase. Data were partitioned 75% into a training dataset (*n* = 1300) and 25% (*n* = 433) into a testing dataset using the ‘tidymodels’ R package [16]. The data splitting process was stratified to ensure that the training and test dataset produced the same frequency distribution of high- and low-risk cohorts in each dataset. Each of the five individual models was created using the training dataset and the final metrics for each model’s performance were obtained using the testing dataset only once. A flow diagram of data preparation, partitioning, and classification is shown in Figure 2.

### 4.4. Recipe Creation

The ‘tidymodels’ R framework was utilized to create a recipe that defines a series of data preprocessing tasks and develops a model specification formula [17]. Within the recipe, BRD morbidity risk (high/low) in the first 45 days on feed was selected as the outcome variable and predictor variables of interest were identified (Table 2). Variables that were not meaningful in external application of the model such as pen ID, cohort ID, and feedlot ID were excluded from analysis. An indicator (or dummy) variable was created for each qualitative variable and converted into a matrix of dummy variables that are 0 or 1 for that categorical variable. This formula and training dataset were used across the five models tested in our analysis.

### 4.5. Classification Algorithms

Five commonly used predictive models were used to predict the BRD risk class of each cohort of cattle. The models used were: logistic regression, decision tree, random forest, naïve Bayes, and linear discriminant. Each individual predictive model was trained with the training dataset. Evaluation of the model performance was performed using the test dataset with the pre-defined cutoff (15%) for BRD morbidity risk within the first 45 days after arrival as the outcome of interest.

Logistic regression is a statistical model used when the outcome variable is binary. It describes the linear relationship between the outcome and the explanatory variables using the logistic function to observe the effect of each variable on the probability of the observed event of interest [18]. The predicted class selected is based on which class has the highest probability. The ‘glmnet’ function in R was used to create the logistic regression models [19].

Decision tree is a hierarchical classification machine learning model composed of decision rules that recursively classify data from the training dataset through a series of questions [20]. Each node in the tree contains a question regarding the predictor variables and question nodes are added incrementally to increase separation of the training data into their categories as effectively as possible [21]. Decision tree models were built using the ‘rpart’ R package [22].

Random forest is a classification machine learning algorithm that generates many classification models and aggregates their results [23]. Random forest models operate as an ensemble that consists of many individual decision trees that arrive at a class prediction. The model’s prediction is determined by the most abundant class. The ‘ranger’ package was used to create the random forest models [24].

Naïve Bayes is a classification algorithm that uses Bayes’ theorem of probability and assumes independence among predictors in a given class [25]. Naïve Bayes models provide a mechanism that uses the training data to estimate the posterior probability of each class given a specific variable. The class with the highest posterior probability is the outcome of the prediction. Naïve Bayes models were built using the ‘naiveBayes’ package [26].

Linear discriminant is a classification algorithm that determines a hyperplane to maximize the separation of the projected means of classification [27]. Groups are specified by the discriminant process and data points are classified by where they lie on the hyperplane. Linear discriminant models were built using the ‘mda’ R package [28].

### 4.6. Resampling/Cross-Validation

A k-fold cross-validation resampling method was applied to the training dataset. The goal of using cross-validation was to generate different versions of the training dataset to estimate how well the models will perform with new data that was not used to train the models. This helps to avoid overfitting and selection bias within each model [15]. In this case, k-fold cross-validation splits the training dataset into k smaller subsets, or folds’, of the data. Each model is trained using k-1 of the folds as training data and the model is validated on the remaining part of the data as a test set. The performance metrics reported by the k-fold cross-validation is the average of the values between all folds of the data [29].

The training data within all five classification models were evaluated with 10 distinct folds. For each iteration, data from the training dataset were randomly partitioned (75% (1300 cohorts out of the original 1733)) into 10 equally sized subsets (folds) of data. The remaining 25% (433 cohorts out of the original 1733) were used as the test dataset. Stratified sampling was done to ensure that each fold had the same frequency distribution of the outcome.

### 4.7. Model Optimization/Tuning

Model optimization/tuning is performed to find a combination of hyperparameters in a given machine learning algorithm that provides the best model performance. Hyperparameters have a direct impact with the model’s learning process and act as model settings that can be adjusted to optimize the model’s performance [29]. A grid search was performed to determine candidate tuning parameter values for each model. Some models have more than one tuning parameter and in this case candidate parameter combinations values are created. The resampling data was used to evaluate each parameter value combination and obtain estimates of how well each candidate model performs. After evaluation, the hyperparameter values that produce the best results in the grid search were selected and used for analysis of the test dataset for final analysis of each model utilizing the cross-fold validation dataset.

### 4.8. Model Evaluation

Final evaluation of the models was performed by allowing each algorithm to classify predictions using the test dataset. Classifier predicted probabilities of BRD morbidity risk of low or high were created for each distinct cohort for each classification model. Receiver-operating characteristic (ROC) curves were created utilizing these probabilities compared with known actual health outcomes using the ‘yardstick’ package in R [16]. ROC curves show the diagnostic ability of binary classification models and the trade-off between sensitivity and specificity for every possible cutoff for a test [30]. The cutoff point that was utilized from each generated ROC curve was the point where sensitivity and specificity were maximized by calculating Youden’s index [31]. Youden’s index has a range between 0 and 1, with the value of 1 indicating the test has perfect sensitivity and specificity. Classification model performance was then evaluated using the final predicted classes based on the cut point selected. Our primary metric for initial model comparison is AUC, because it is a measure of the degree of separability and how well the model can distinguish between classes using a range from 0 to 1, where a value of 0 indicates a perfectly inaccurate test and a value of 1 indicates a perfectly accurate test [32]. Additional metrics calculated and evaluated were true positives (TP), true negatives (TN), false positive (FP), false negatives (FN), Positive Predictive Value (PPV), Negative Predictive Value (NPV), sensitivity (Se), specificity (Sp), and accuracy. Figure 3 displays a flowchart describing how a model would arrive at each diagnostic outcome and the calculations for each metric.

### 4.9. Economic Analysis

An economic analysis was performed with the goal estimating a cohort-level Net Health Cost Benefit (NHCB) for each predictive model and a control scenario that represented a person classifying expected BRD risk without the use of a model. This NHCB was meant to represent the health costs associated with each model to predict expected BRD risk. These health costs include expenses associated with the administration of BRD treatment and the potential lost value from a morbid animal compared to a healthy animal. The values for these costs were determined based on previous reports and averages from the study population dataset (Table 3). BRD morbidity was defined as the number of cattle within a cohort that were treated for respiratory disease at least once in the first 45 DOF. The cost of a morbid animal was considered as $151.18 per head based on data from a previous Texas A&M Ranch to Rail summary report [33]. This cost considered the return difference from healthy animals compared to sick animals in medicine costs and ‘lost value’ due to reduced efficiency, lowered gain, and reduced sale value. The cost of a single metaphylaxis treatment ($23.60) was the average cost for a respiratory disease treatment reported in a USDA NAHMS report [34]. The average cohort size (*n* = 109) was set as the average cohort size in our study population dataset and was calculated by taking the total number of animals in our study population and dividing it by the total number of cohorts in the population (*n* = 188,118/1733). The proportion of high-risk cohorts served as a range that represented the proportion of high-risk cohorts that could have potentially been present in the data. For example, if this number was set as 0.25 then 25% of the cohorts in the dataset would be expected to be high risk for BRD. The metaphylaxis efficacy was set at 0.5 (50%) to represent the reduced morbidity after metaphylaxis treatment [35].

In this analysis there was a decision on whether an incoming cohort is going to receive or not receive metaphylactic treatment; thus, a cohort could be true positive (had ≥15% morbidity within the first 45 DOF and were predicted to be high risk) or a false positive (predicted to be high risk and had <15% morbidity within the first 45 DOF). In our analysis we varied the prevalence of high-risk cohorts (0–100) to represent feedlots with different types of incoming cattle. The total number of diagnostic outcomes (TP, TN, FP, FN) were calculated at each level of prevalence for each predictive model. A cost incorporating the cost of metaphylaxis and lost value from a morbid animal was assigned to each diagnostic outcome (TP, TN, FP, FN) to calculate the NHCB. The NHCB was then subsequently divided by the average cohort size of animals in the study population (mean = 109) to produce an average cost/benefit per animal for use of each model. Table 4 describes the potential diagnostic outcomes generated from each model prediction with the associated metaphylaxis decision that would be decided by feedlot management and the anticipated financial result from the decision made. The NHCB was then subsequently divided by the average cohort size of animals in the study population (mean = 109) to produce an average cost per animal for use of each model. The formulas for the NHCB and cost of each diagnostic outcome are shown below:NHCB = TPcost + TNcost + FPcost + FNcost(1)
TNcost = $0 (baseline; no incurred costs)(2)
TPcost = (MC × (BRD45 × ME) × TC) − (CS × TC)(3)
FPcost = −(TC × CS)(4)
FNcost = −((BRD45 × ME) × MC × CS)(5)

TNcost, TPcost, FPcost, and FNcost represent the cost per animal of a true negative, true positive, false positive, and false negative outcome, respectively. MC represents the cost incurred from the lost value of a morbid animal compared with a healthy animal ($151.18). BRD45 represents the percent of cohorts in the population that are expected to be high risk for BRD morbidity in the first 45 DOF and in our analysis could take a value from 0 to 100. ME represents the metaphylaxis efficacy, estimated to be a 50% reduction in morbidity. TC represents the average treatment cost for a single metaphylactic treatment ($23.60). CS represents the average size of a cohort (109).

A control scenario was included to represent a human classifying expected BRD risk to incoming cohorts without the use of a predictive model. This was added in order to compare the cost of using a model against standard methods to predict expected BRD risk. A NHCB for a control scenario was also calculated to compare the economic output between the model results and the control scenario. To achieve this, Se and Sp were calculated from a subset of data (*n* = 177 cohorts) that included the actual risk status assigned to each cohort by feedlot management. This Se and Sp were calculated by comparing the feedlot’s classifications with the actual health outcomes for each cohort based on the 15% BRD morbidity cutoff that was used in the modeling process. For example, if a feedlot classified an incoming cohort as high risk and the percent of the cohort that was treated for BRD once was 15% or greater, then it was called a TP. Diagnostic outcomes (TP, FP, TN, and FN) were calculated based on these criteria. The NHCB was formulated in the same manner as the costs for each model using the calculated sensitivity and specificity. A difference from the control ($/animal) was calculated at each proportion of high-risk cohorts to low-risk cohorts to compare the NHCB between the five models and the control. The control was set at $0/animal and all models were compared with the control. If a model displayed a value greater than $0/animal at any proportion of high-risk cohorts then that indicated that there was a potential economic advantage to use the model relative to the control. If a model displayed a value less than $0/animal at any proportion of high-risk cohorts then that indicated that there was a potential economic disadvantage to use the model relative to the control.

## 5. Conclusions

The objectives of this study were to evaluate the diagnostic performance of five classification models to classify incoming groups of cattle into high- and low-risk categories based on the BRD morbidity within the first 45 DOF and evaluate the models using an economic framework to determine whether the models were advantageous to a person classifying expected risk. We used AUC to evaluate model performance as this metric measures the models’ degree of separability between high- and low-risk cohorts. Using area under the curve, the random forest model had the best performance with a value of 0.789 using the testing dataset. Although the random forest model had the highest AUC, it was not always the best model to use economically. The economic performance of each model was dependent on the prevalence of high-risk cohorts in the population. The decision tree provided the greatest estimated economic benefit when the proportion of high-risk cohorts was lower than 45% in the population. In addition to previously evaluated factors, this study provides a new outlook using arrival and pen housing factors to classify cohorts into risk categories. In order to further evaluate the true impact of these predictive models a prospective study should be considered to validate the true diagnostics and costs of using a predictive algorithm compared with current management strategies to determine the expected BRD risk of incoming cohorts of cattle. In addition, more data, including new predictor variables and observations of data, are needed to continue to refine the algorithms and provide a better estimate of each model’s predictive performance.

## Figures and Tables

**Figure 1 pathogens-11-00442-f001:**
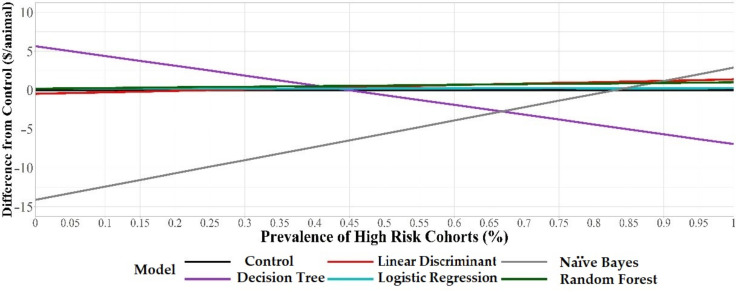
Estimated economic results ($/animal) of five classification models compared with a person (control) classifying expected BRD morbidity risk of incoming cattle cohorts in the first 45 DOF across varying proportions of high-risk cohorts to low-risk cohorts (0–100%). Gray line represents the control ($0/animal). At any prevalence, if the difference from the control for a model is above the control line ($0/animal) then it has a potential economic advantage relative to the control. If the difference from the control for a model is below the control line ($0/animal) then it has a potential economic disadvantage relative to the control.

**Figure 2 pathogens-11-00442-f002:**
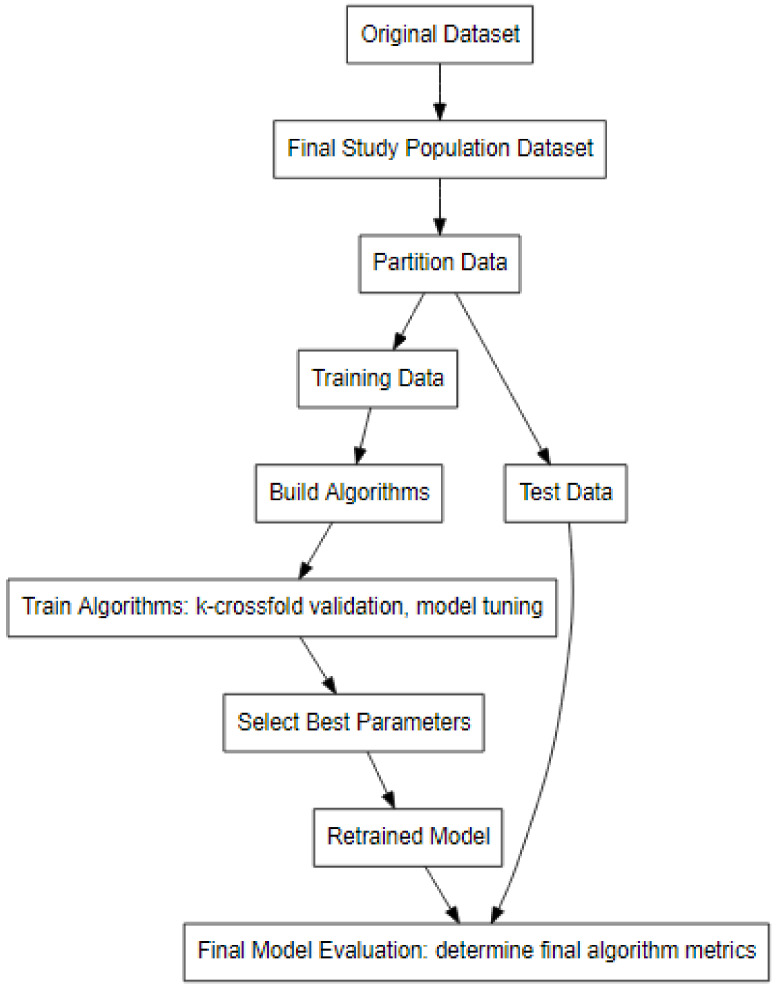
Flowchart of data refinement, data partitioning, algorithm training, and classification model algorithm evaluation.

**Figure 3 pathogens-11-00442-f003:**
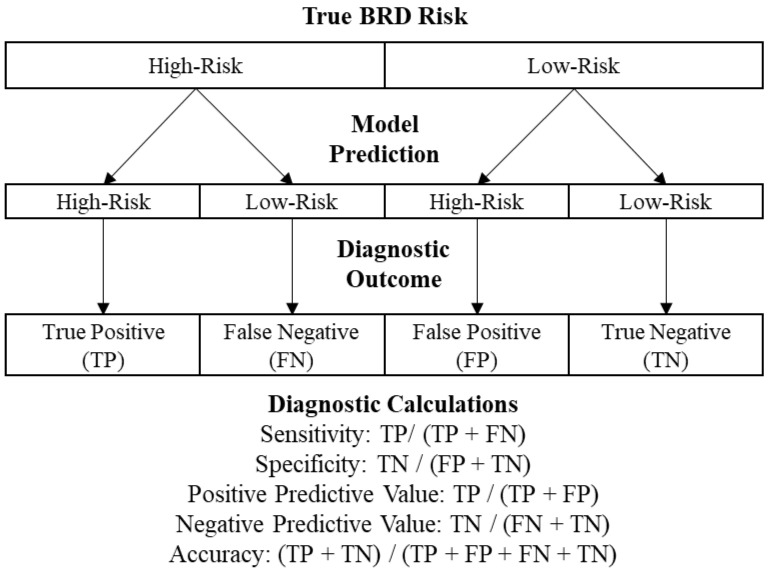
Flowchart of diagnostic outcomes and calculations generated from predictive classification models using cutoff of 15% BRD morbidity in the first 45 DOF.

**Table 1 pathogens-11-00442-t001:** Final diagnostic performance estimates utilizing the test dataset for BRD morbidity risk during the first 45 days on feed to classify cohorts as high or low risk for BRD development within the first 45 days post arrival using a 15% cutoff at the optimum cutoff where 10.85% of the cohorts had >15% BRD morbidity.

Performance Metric	Logistic Regression	Decision Tree	Random Forest	Naïve Bayes	Linear Discriminant
AUC ^1^	0.785	0.682	0.789	0.743	0.760
True Positives	40	21	42	47	37
False Positives	152	63	153	386	141
True Negatives	234	323	233	0	245
False Negatives	7	26	5	0	10
Accuracy%	63.3	79.4	63.7	10.9	61.4
Sensitivity%	85.1	44.7	89.4	100.0	91.5
Specificity%	60.6	83.7	60.6	0.0	57.8
PPV% ^2^	20.8	25.0	21.3	10.9	20.8
NPV% ^3^	97.1	92.6	97.9	DBZ ^4^	96.1
AUC ^3^	0.785	0.682	0.789	0.743	0.760

^1^ AUC—Area under the curve. ^2^ PPV—Positive Predictive Value. ^3^ NPV—Negative Predictive Value. ^4^ DBZ—Division by zero (error).

**Table 2 pathogens-11-00442-t002:** Description of predictor and outcome variables.

Variable	Description
Cohort size at arrival	Total animals in cohort upon arrival to the feedlot
Average arrival weight at arrival	Total weight of all animals/cohort size at arrival
Arrival Date Quarter ^1,2^	Quarter of the year that cohort arrived (1,2,3,4)
Sex ^1^	Gender of the cohort (steer, heifer, mixed gender)
Total pen area (sq. m)	Total area of the pen that cohorts were placed in
Bunk space length (m)	Total length of bunk available in pen
Pen area available per head (sq. m)	Total pen area/cohort size at arrival
Bunk space available per head (m)	Bunk space length/cohort size at arrival
BRD morbidity risk ^3^	1 = total cohort BRD morbidity risk ≥ 15% 0 = total cohort BRD morbidity risk < 15%

^1^ Qualitative variables that were converted to quantitative variables as dummy variables. ^2^ 1 (January, February, Mar), 2 (April, May, June), 3 (July, August, September), 4 (October, November, December). ^3^ Binary outcome variable.

**Table 3 pathogens-11-00442-t003:** Variables included in the economic analysis to compare the cost benefit of using one of the predictive models compared with the control scenarios.

Variable	Value
Total number of lots ^1^	1733
Average cohort size ^2^	109
Cost of single metaphylactic treatment per animal ^3^	$23.60
Prevalence of high-risk cohorts (%)	0–100
Cost of morbid animal ^4^	$151.18
BRD morbidity% in true positives ^5^	27%
BRD morbidity% in true negatives ^6^	3%
Metaphylaxis efficacy ^7^	0.5

^1^ Total of number of cohorts in study population. ^2^ Average cohort size in study population. ^3^ Average cost per animal to administer metaphylaxis (USDA 2013). ^4^ Average cost of a sick animal [33]. ^5^ Average morbidity in the true positive (high-risk) cohorts in study population. ^6^ Average morbidity in the true negative (low-risk) cohorts in study population. ^7^ Metaphylaxis efficacy set at 0.5 (50%) to represent reduced morbidity after metaphylaxis treatment [35].

**Table 4 pathogens-11-00442-t004:** Possible diagnostic outcomes associated with an economic analysis to evaluate the costs associated with correct or misclassification of cohort-level risk of bovine respiratory disease in first 45 days on feed.

Diagnostic Outcome	Truth	Model Prediction	Metaphylaxis Decision	Financial Consequence
TP	High risk	High risk	Treat	Animals that are truly high risk are metaphylactically treated. Treatment costs are incurred, but expenses are saved by avoiding lost value from potential morbid animals.
FN	High risk	Low risk	Do not treat	Animals that are truly high risk are not treated. These animals are expected to become morbid and provide lower value compared to healthy animals. The magnitude of financial loss was dependent on the prevalence of high-risk cohorts.
FP	Low risk	High risk	Treat	Animals that are truly low risk are metaphylactically treated. These animals are expected to be healthy, but received treatment regardless, so the incurred costs are only the treatment cost of metaphylaxis for animals in each cohort.
TN	Low risk	Low risk	Do not treat	Animals that are truly low risk are not metaphylactically treated. This is the baseline cost that was compared with all other outcomes. Since animals are expected to be healthy, and no treatment costs are incurred, the value for this outcome will always be $0.

## Data Availability

Data used in this study were from cooperating entities and are not available publicly due to confidentiality and anonymity agreements.

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
