# Peer review of "Predicting Bovine Respiratory Disease Risk in Feedlot Cattle in the First 45 Days Post Arrival"

_pathogens, 2022, doi:10.3390/pathogens11040442_

Round 1
Reviewer 1 Report
The Editor Pathogens Thank you for the opportunity to review the manuscript: “Predicting bovine respiratory disease risk in feedlot cattle in 2 the first 45 days post arrival”. The paper has been carefully reviewed but significant concerns arose:
More references should be added in the introduction, the intention to place the reader the importance of the problem. The discussion were very extensive and confusing, with repetition of information . Figure 1 must be improved. However, the work has great importance, presenting important scientific merits.
Author Response
Response: we appreciate your feedback on the manuscript and have tried to adjust the manuscript in response to your comments. Figure 1 is now included as a color figure and we will defer to the editor for other potential suggestions to improve the presentation of this information. We have edited the rest of the manuscript to improve clarity and reduce repetition. Thank you for your feedback on this manuscript.
Reviewer 2 Report
This paper considers models to assess the risk of morbidity in feedlot cattle due to bovine respiratory disease, using retrospective data and known risk factors. The paper is well-written, a high standard of English has been used throughout, and the article is well-structured with clear use of subheadings to lead the reader. The study is novel in its approach, but also includes a very large data set. Data was obtained from numerous feedlots and represents a range of background conditions and levels of morbidity, albeit only from one large region, but is likely to be representative of conditions commonly encountered, such that the results are generalisable. Identification of the economic benefit of different models was an important novel aspect of this study, since margins in feedlots are typically low, and improved means of identifying optimum practice will increase margins and cattle welfare. The fact that the benefit of each model differed with the prevalence of high risk cohorts was an important finding - supporting the conclusion that more work needs to be done to assess the practical benefit of predictive tools compared with current experience-based management. The authors could consider the role of further education of feedlot managers to better identify at-risk cohorts or individual cattle, and appropriate treatment. I wonder how to determine which model to apply in practice, since the optimal model depended on the level of morbidity, which would be unknown at the time of arrival at feedlots.
Author Response
Response: thank you for your consideration and input on the review. You are correct, the data was gathered from one geographic region; however, this region represents most of the fed cattle in the US. Therefore, we believe the results are applicable to a variety of operations. We also appreciate the understanding of relatively low margins in the industry and results may be valuable to operations. We will take into account your input when we are sharing results of work relative to discussing with feedyard managers which model is most appropriate for their operation dependent on the expected level of morbidity.
Reviewer 3 Report
Dear authors,
More information is needed on eligible criteria of the animals participating in the cohorts.
Overall, the study is well documented and presented.

Author Response
Response: We appreciate your input on the manuscript and have made adjustments in response to your comments. Unfortunately we did not have any information available on cattle management or preventative health procedures prior to arrival at the feedyard and only were able to utilize data collected from the feedyard on each cohort. We have modified the manuscript to include this information.